# Application of Machine Learning in Hospitalized Patients with Severe COVID-19 Treated with Tocilizumab

**DOI:** 10.3390/jcm11164729

**Published:** 2022-08-12

**Authors:** Antonio Ramón, Marta Zaragozá, Ana María Torres, Joaquín Cascón, Pilar Blasco, Javier Milara, Jorge Mateo

**Affiliations:** 1Department of Pharmacy, General University Hospital, 46014 Valencia, Spain; 2Institute of Technology, University of Castilla-La Mancha, 16002 Cuenca, Spain; 3Department of Pharmacology, Faculty of Medicine, University of Valencia, 46010 Valencia, Spain; 4Centre for Biomedical Research Network on Respiratory Diseases (CIBERES), Health Institute Carlos III, 28029 Madrid, Spain

**Keywords:** COVID-19, SARS-CoV-2, machine learning, cytokine release syndrome, tocilizumab

## Abstract

Among the IL-6 inhibitors, tocilizumab is the most widely used therapeutic option in patients with SARS-CoV-2-associated severe respiratory failure (SRF). The aim of our study was to provide evidence on predictors of poor outcome in patients with COVID-19 treated with tocilizumab, using machine learning (ML) techniques. We conducted a retrospective study, analyzing the clinical, laboratory and sociodemographic data of patients admitted for severe COVID-19 with SRF, treated with tocilizumab. The extreme gradient boost (XGB) method had the highest balanced accuracy (93.16%). The factors associated with a worse outcome of tocilizumab use in terms of mortality were: baseline situation at the start of tocilizumab treatment requiring invasive mechanical ventilation (IMV), elevated ferritin, lactate dehydrogenase (LDH) and glutamate-pyruvate transaminase (GPT), lymphopenia, and low PaFi [ratio between arterial oxygen pressure and inspired oxygen fraction (PaO_2_/FiO_2_)] values. The factors associated with a worse outcome of tocilizumab use in terms of hospital stay were: baseline situation at the start of tocilizumab treatment requiring IMV or supplemental oxygen, elevated levels of ferritin, glutamate-oxaloacetate transaminase (GOT), GPT, C-reactive protein (CRP), LDH, lymphopenia, and low PaFi values. In our study focused on patients with severe COVID-19 treated with tocilizumab, the factors that were weighted most strongly in predicting worse clinical outcome were baseline status at the start of tocilizumab treatment requiring IMV and hyperferritinemia.

## 1. Introduction

COVID-19 is an acute respiratory illness caused by severe acute respiratory coronavirus-2 (SARS-CoV-2) [1]. Since December 2019, it has caused a pandemic, affecting millions of lives worldwide. It presents a clinical course in two phases. The first is an asymptomatic or silent viral replication phase of about five days after infection, with symptoms of cough, fever and dyspnea. This phase may progress to a second more severe phase that can lead to bilateral pneumonia with severe respiratory failure (SRF) with or without acute respiratory distress syndrome (ARDS) and/or fatal multiorgan failure [2,3]. About 10% of patients require care in intensive care units (ICU) [4]. The mortality rate in the first wave was <3%, although the case fatality rate of severe cases was high, according to the World Health Organization (WHO).

Risk factors associated with the progression and severity of SARS-CoV-2 have been identified as advanced age, different comorbidities, as well as laboratory parameters, such as lactate dehydrogenase (LDH), procalcitonin, high-sensitivity C-reactive protein (CRP), ferritin, and proinflammatory cytokines such as interleukin (IL)-6 and IL-1β [5,6]. Elevated IL-6 levels and hyperferritinemia are considered indicators of systemic inflammation and poor prognosis in COVID-19 [7]. The neutrophil-to-lymphocyte ratio is usually elevated in patients with severe COVID-19 [8]. Potential complications include acute kidney injury, coagulation disorders, ARDS, or shock that may result in mortality rates >30% [5].

In patients with severe COVID-19, a hyperinflammatory response to SARS-CoV-2 occurs due to a dysregulated innate host immune response, with increased levels of proinflammatory cytokines (IL-1, IL-6, TNF-α) and other acute phase reactants (CRP, D-dimer or ferritin) [9]. This cytokine release syndrome (CRS), or “cytokine storm”, plays a key role in the pathogenesis of severe COVID-19, including lung damage and microvascular thrombosis [10], potentially leading to multiorgan failure and death [4]. The IL-6 signaling pathway is one of the key mediators of the hyperinflammatory syndrome and CRS in COVID-19. IL-6 is involved in the signal transduction pathway through Janus kinases-signal transducers and activators of transcription (JAK-STAT). This pathway is essential in the processes of cell proliferation and differentiation, immunity and apoptosis [3,4]. Blocking this pathway at different levels could prevent dysregulation of the inflammatory, immune and coagulation systems and limit lung damage, reducing the risk of progression to SRF. For this reason, current research is focused on the use of different molecules capable of blocking the cytokine storm, such as IL-1 or IL-6 antagonists or JAK-STAT inhibitors.

Tocilizumab is a humanized monoclonal antibody which binds to the soluble and membrane-bound IL-6 receptor, blocking the signal transduction by which JAK-STAT is activated, perpetuating the cytokine storm [11]. Tocilizumab has demonstrated a favorable benefit–risk ratio, improving the prognosis of patients with COVID-19 [4,12,13]. The recent meta-analysis by Luo L et al. concludes that tocilizumab significantly reduces the relative risk of 28–30-day mortality and the risk of invasive mechanical ventilation (IMV), without increasing the risk of infection and/or adverse events [14].

Based on all these data, the European Society of Clinical Microbiology and Infectious Diseases suggests using tocilizumab in patients with SARS-CoV-2-related SRF [15]. Despite this evidence, the risks of mortality and IMV in patients treated with tocilizumab are not negligible [15]. Currently, little is known about the predictive factors associated with poor outcome in patients with COVID-19 treated with immunomodulators in general and tocilizumab in particular in clinical practice.

To solve this lack of knowledge, machine learning (ML) models can be used, designed to make accurate predictions on any question of interest, using data from a multitude of variables, as opposed to classical statistical models created to make inferences about the relationships between variables. ML, as part of artificial intelligence (AI), uses statistical and mathematical algorithms that enable it to opt for patterns that help to make complex decisions [16]. These algorithms can be used to develop predictive models and reduce the complexity of clinical phenotypes. AI tools have been implemented in the fight against COVID-19. They have been used primarily in: early diagnosis; the prediction of severity, mortality and complications; as well as in drug and vaccine discovery or repurposing [17,18,19].

To our knowledge, this is the first study to develop, compare and validate five ML models to predict factors associated with reduced clinical benefit of tocilizumab, in terms of mortality and hospital stay, in patients with SRF due to SARS-CoV-2.

## 2. Materials and Methods

### 2.1. Data Source

Patient data were obtained from different internal hospital sources: (1) electronic medical record (EMR) (Hosix. Net. Ink.), which includes a module for recording clinical analysis results and an electronic drug prescription module, and (2) ICU prescription software (IntelliSpace Critical Care and Anesthesia, version H.02.00. Philips Ibérica SA, Eindhoven, The Netherlands). With this information, an individualized data collection questionnaire (DCQ) was constructed for each patient.

### 2.2. Study Design and Population

This was a longitudinal, retrospective, observational study conducted in a tertiary hospital. Sixty-seven patients (64.2% male) admitted to the hospital with microbiologically confirmed SARS-CoV-2 by reverse transcriptase polymerase chain reaction (RT-PCR) assay of a nasopharyngeal swab between 10 March and 16 September 2020 were included.

Inclusion criteria were: patients > 18 years old admitted to the General University Hospital of Valencia (GUHV) with severe COVID-19 pneumonia, with comorbidity and requiring admission to the ICU. The patients selected had to be candidates for tocilizumab administration according to the hospital’s internal protocol at the start date of the study: onset of symptoms ≥ 7 days, interstitial pneumonia with SRF and extensive pulmonary infiltrate, rapid respiratory worsening requiring noninvasive or invasive ventilation (COVID respiratory severity scale ≥ 3), presence of other organ failure (mainly shock or SOFA scale score ≥ 3) and criteria for severe systemic inflammatory response: elevated IL-6 levels (>40 pg/mL) or alternatively elevated D-dimer levels (>1500 ng/mL) or progressively increasing D-dimer. The approved posology for tocilizumab was: (a) patients ≥ 80 kg, who were given a first dose of 600 mg in 100 mL of sterile, non-pyrogenic sodium chloride 9 mg/mL (0.9%) solution for injection and 1-h intravenous infusion and a second dose of 600 mg after 12 h of the first dose. In case of persistent fever and worsening of analytical parameters (CRP, D-dimer or IL-6) a third dose could be considered 16–24 h after the second dose; and (b) patients < 80 kg, who were given a first dose of 600 mg and a second dose of 400 mg. Likewise, a maximum of 3 infusions could be considered, only if there is no significant clinical improvement.

In our study, all patients received corticosteroids: (a) methylprednisolone 250 mg/24 h (1 day) followed by 40 mg/12 h (4 days) intravenously, or (b) dexamethasone 6 mg/24 h for 10 days (starting intravenously and switching to oral if clinical stability and tolerance). Similarly, all patients received azithromycin 500 mg/24 h (5 days) and prophylactic doses of enoxaparin (60 mg/24 h if BMI < 30 or 80 mg/24 h if BMI ≥ 30). If D-dimer > 2000 ng/mL, enoxaparin 1.5 mg/kg/24 h.

Exclusion criteria were: patients ≤ 18 years, patients with missing data for more than one clinical/analytical variable in this period, and patients who were not candidates for tocilizumab for any of the following reasons: GOT and/or GPT > 5 times normality levels, neutrophils < 500 cells/mm^3^, platelets < 50,000 cells/mm^3^, documented sepsis due to pathogens other than SARS-CoV-2, complicated diverticulitis or intestinal perforation, ongoing skin infection (e.g., pyodermitis not controlled with antibiotic treatment), immunosuppressive anti-rejection therapy, and/or use of monoclonal antibodies that could produce immunosuppression.

The study was approved by the Ethics Committee of the GUHV. Participants gave informed consent before participating in the study. Protocol code: XGB-COVID-19.

### 2.3. Study Data

Data on demographic, clinical and laboratory variables, as well as the date of hospital admission, were included in the DCQ. In all patients, data on laboratory parameters were collected at the moment of tocilizumab initiation. Demographic variables included age and sex, weight and height. Clinical variables included: (1) pharmacotherapy prior to and on admission: (a) use of antivirals: oral lopinavir/ritonavir (400/100 mg/12 h) or intravenous remdesivir (200 mg on day 1, followed by 100 mg/day for up to 10 days); (b) use of oral hydroxychloroquine [400 mg/12 h (day 1) and 200 mg/12 h (days 2–4)]; (c) use of subcutaneous interferon beta-1b 250 mcg/48 h; (d) use of subcutaneous anakinra, preferably in patients with poor evolution (PaO_2_/FiO_2_ < 300, SpO_2_ < 92% in room air, tachypnea or high ferritin). Doses of 200 mg/24 h (day 1) and 100 mg/24 h (days 2–5); (2) baseline patient status at initiation of tocilizumab therapy: (a) patient requiring supplemental O_2_ but not IMV and (b) patient requiring IMV; (3) clinical status at day 7 and 21 after hospital admission: (a) hospital discharge with resumption of normal activities; (b) hospital discharge with difficulty resuming normal activities; (c) hospitalized not requiring supplemental O_2_; (d) hospitalized requiring supplemental O_2_ but not IMV; (e) hospitalized requiring IMV and (f) *exitus*; (4) comorbidities: smoking, arterial hypertension, obesity, ischemic heart disease, dyslipidemia, diabetes mellitus (DM), chronic obstructive pulmonary disease (COPD) and chronic kidney disease (CKD); and (5) use or not of IMV and types. Finally, the following laboratory variables were collected: PaO_2_/FiO_2_, FiO_2_, SpO_2_, CRP, ferritin, LDH, glutamate-oxaloacetate transaminase (GOT) and glutamate-pyruvate transaminase (GPT), procalcitonin and lymphocytes.

To measure the efficacy of tocilizumab, in-hospital mortality data and days of hospital stay were recorded.

### 2.4. Method

#### 2.4.1. Model Development

For this study, the extreme gradient boost (XGB) based algorithm has been proposed as the reference algorithm because it presents several important features such as its scalability, parallel computing and its speed in execution [20,21]. To build this classification model we start from a dataset *S* = *x_j_*, *y_j_*,
(1)$$\hat{y_{j}}=\overset{P}{\underset{p=1}{\sum }}t_{p}\left(x_{j}\right)$$
where *x_j_* represents the input vector with *m* time variables, $\hat{y_{j}}$ shows the predicted output, *y_j_* is the output, *t_p_* represents a tree with leaf weight *w_p_* and structure *u_p_*, j = 1; 2; ...; *n*, and *P* corresponds to the number of trees.

The regularized objective function is presented in Equation (2) for the proposed XGB method. A second-order Taylor expansion is applied to approximate the XGB objective function in order to optimize the prediction accuracy [20].
(2)$$R=\underset{j}{\sum }r\left(\hat{y_{j}},y_{j}\right)+\underset{p}{\sum }\Psi \left(t_{p}\right)$$
(3)$$\Psi \left(t_{p}\right)=\lambda f_{p}+\frac{1}{2}\gamma ∥\omega_{p}∥{}^{2}$$

In Equation (3), *f_p_* shows the number of leaves on the tree. The function *R* penalizes the complexity of the method. The learning rate is shown by *λ*, where *w_p_* is the vector of leaf scores. By means of the parameter *γ* the complexity of the system is monitored. The objective is to optimize Equation (2) [21].

In this work, other ML algorithms have been implemented to test the performance of the proposed method. The four that gave the best results in the comparison were selected. The following systems stood out: decision tree (DT) [22], Bayesian linear discriminant analysis (BLDA) [23], K-nearest neighbour (KNN) [24], and support vector machine (SVM) [25]. All of them were designed using Matlab 2022 software. For the validation of the results, a 5-fold cross-validation was performed to avoid overfitting. The input data was divided into two blocks, 70% was used for training and 30% for validation. In all simulations, 100 iterations were performed to obtain the standard deviation and mean values in a uniformly random manner. In this way, the impact of noise is reduced, appropriate values are calculated and statistically valid results are obtained [26]. In Figure 1, the procedure carried out in this study can be seen.

#### 2.4.2. Performance Evaluation

In this work, the following metrics were used to test the performance of the compared methods: degenerated Younden index (DYI), specificity, recall, precision, balanced precision, receiver operating characteristic (ROC), F1-score, Matthew’s correlation coefficient (MCC), Cohen’s Kappa index (CK), and area under the curve (AUC) [26].

## 3. Results

This section describes the results obtained with the patient registries used for training and validation to define the factors with the greatest influence on in-hospital mortality and days of hospital stay in patients with severe COVID-19 treated with tocilizumab. The performance of the proposed system has been compared with different ML classification methods accepted in the scientific community.

Table 1 presents the results obtained from the classification methods applied: SVM, DT, BLDA, KNN and the proposed XGB system. As can be seen, the systems based on SVM and BLDA obtain a lower accuracy value than the rest of the methods; these values are close to 82%. As for the DT and KNN methods, they improve the classification capacity obtaining accuracy values of 86%, presenting a better performance than SVM and BLDA. On the other hand, the proposed XGB system achieves accuracy values of 93%, a significant increase over previous methods, which translates into better prediction. As can be seen in Table 1, the same happens again with the F1 score parameter, where XGB obtains higher values that imply an improvement in classification.

To test the performance of the proposed XGB system, other parameters widely used in the literature were calculated, such as the AUC, the MCC, the DYI and the Kappa index. For this analysis, one of the most reliable statistical indices available, the MCC, was used. This coefficient produces a high score only if the prediction has performed well in all four categories of the confusion matrix (true positives, false negatives, true negatives and false positives). The results in the four categories of the matrix are proportional to the size of the positive elements and the size of the negative elements in the dataset. As can be seen in Table 1, the proposed method XGB achieves a value of 84.41%, increasing the values achieved by KNN and DT. The latter show values of 76.97% and 74.18%. Both SVM and BLDA show a worse performance in this parameter. As for the Kappa index, XGB obtains a value close to 84%, improving the value of KNN and DT by 6.76% and 10.02%, respectively. As for the AUC and DYI parameters, the same is true again; the XGB method achieves higher values, which means that it obtains a better classification of patients who may benefit less from tocilizumab treatment.

On the other hand, the ROC curve has been used to compare the classification ability of the proposed system with the other ML methods. The curve is the result of plotting the sensitivity and specificity for each threshold value [26]. As can be seen in Figure 2, the results obtained by the different systems for classification between patients according to the objective of the proposed study, are shown. A larger surface area is seen for the XGB method, which implies a better classification of the two classes; these values can be seen in Table 1.

For clarity, all metrics have been grouped for each data set (training and validation) and are presented as a radar plot. A perfect score on all metrics would be represented by a circle the size of the entire grid. In our study, the training sets of the model have higher scores on all metrics and, in general, have lower scores on the validation set. The shape of the graphs can also be indicative of the quality of the models. The larger the area of the circle in the validation set, the better the prediction method. The proposed XGB system (Figure 3) is a good example of a balanced model. The training and validation sets result in similar pie charts. These similarities are because the system obtains an optimal training point, with no overfitting or underfitting, making the method highly generalisable. That is, given a new input, the system does well to provide a correct output. As can be seen, the BLDA method was the worst performer on most metrics. In view of the results obtained, we can say that the proposed XGB system manages to classify patients according to the study objective with high accuracy and automatically, which makes this tool very helpful for clinical practice.

The parameters used for the implementation of the different methods are shown in Table 2.

With the proposed XGB method, the factors associated with a worse outcome of tocilizumab use in terms of mortality were: baseline situation at the start of tocilizumab treatment requiring IMV, elevated ferritin, LDH and GPT, lymphopenia, and low PAFI values. The factors associated with a worse outcome of tocilizumab use in terms of hospital stay were: baseline situation at the start of tocilizumab treatment requiring IMV or supplemental oxygen, elevated levels of ferritin, GOT, GPT, CRP, LDH, lymphopenia, and low PAFI values.

The baseline clinical data of the 67 patients included in the study are shown in Table 3.

## 4. Discussion

To our knowledge, this is the first study to develop, compare and evaluate five supervised ML methods to predict the factors that most negatively influence the efficacy, in terms of mortality and hospital stay, of tocilizumab in patients hospitalized with SRF due to SARS-CoV-2. Forty-one demographic, clinical and laboratory variables were collected. After analysis of the five ML algorithms, the XGB method had the highest balanced accuracy (93.16%).

Infection of airway epithelial cells by SARS-CoV-2 causes cytological damage by activating the local immune response, releasing proinflammatory mediators such as IFNγ, IL-1β, IL-6, TNF-α [27]. IL-1β acts on endothelial and vascular smooth muscle cells and induces IL-6 production [28]. IL-6, through induction of proinflammatory chemokines and cytokines, plays a critical role in the progression from mild inflammation to hyperinflammatory conditions, CRS, ARDS and corresponding lung damage causing mortality in critically ill COVID-19 patients [29,30]. Therefore, blockade of the IL-6 pathway is considered a key approach to reducing severe lung damage caused by SARS-CoV-2 [4]. ARDS is the most common cause of death in these patients, with mortality rates that can exceed 75% [3,31]. ARDS is a consequence of dysregulated immune response, hyperinflammation and activation of the coagulation cascade, with CRS playing the major role [32]. Currently available treatment focuses on modulating immune responses, and obtaining antiviral, antithrombotic or anticoagulant effects [3]. The drugs with the most accumulating evidence for CRS blockade are IL-1 (anakinra) or IL-6 (tocilizumab or sarilumab) inhibitors, janus kinase inhibitors [JAKi] (e.g., baricitinib), as well as corticosteroids [4,33,34]. Increasing evidence supports the incremental efficacy of corticosteroids alone or in combination with tocilizumab/JAKi in moderate to severe and critical COVID-19 [4,35,36]. Therapeutic plasma exchange also appears to have potential benefits in reducing the risk of mortality among critically ill COVID-19 patients [37]. Of this therapeutic arsenal, tocilizumab is the most studied drug in the setting of severe COVID-19. Most randomized clinical trials (RCTs) and observational studies show favourable evidence that tocilizumab, when added to standard therapy, significantly reduces both 30-day mortality and the need for IMV without causing serious adverse events [38,39,40,41]. The RECOVERY study, initiated in March 2020, is the RCT with the largest number of participants enrolled so far, to test potential treatments against COVID-19 [38]. More than 4000 hospitalized COVID-19 patients with hypoxia and systemic inflammation were included. Tocilizumab was found to improve survival and other clinical outcomes, such as time to hospital discharge and the number of patients eventually receiving IMV. These benefits were observed regardless of the amount of ventilatory support and were in addition to the benefits of systemic corticosteroids (82% of patients received baseline corticosteroids). Despite this favourable evidence, a non-negligible percentage of patients with SRF due to SARS-CoV-2 treated with tocilizumab ultimately dies or undergoes IMV. Indeed, a recent meta-analysis of RCTs and cohort studies including more than 3000 patients with COVID-19 on tocilizumab treatment showed that 30-day mortality was more than 20%, mainly in the most severe patients, similar to the incidence of IMV [14]. Therefore, studies to identify factors influencing response to tocilizumab are warranted.

In our study, in-hospital mortality was 35.8% and 19.4% of all patients required IMV. The median hospital stay was 14 days (interquartile range [IQR = 22]). Of all the ML classifiers applied, the XGB method was the pattern recognition method that most accurately discriminated between patients at increased risk of poor outcome with tocilizumab use from those who were not. This model was analyzed and compared with different supervised ML methods described in the literature, such as BLDA, DT, KNN or SVM. In our study, BLDA and SVM were the worst performing methods, with KNN being the method that comes closest to the accuracy values of XGB. This is consistent with the results of studies describing these supervised ML algorithms in predicting COVID-19 mortality [42,43]. Our study provides a balanced radar graph between the training and test phases. The results show that the XGB model can handle large data dimensions while avoiding overtraining and significantly improves the performance of other classification methods. It obtains higher precision, recall and accuracy values than those achieved by the other methods. This guarantees its reliability for automatic classification of the desired result. XGB is a predictive model that has excellent scalability and high execution speed, used in various fields of biomedicine, such as in predicting the stage of cancer patients [44].

In our research, the factors associated with a worse outcome of tocilizumab in terms of mortality were: baseline status at initiation of tocilizumab treatment requiring IMV, elevated ferritin, LDH and GPT, lymphopenia, and low PaFi [ratio of arterial oxygen pressure to inspired oxygen fraction (PaO_2_/FIO_2_)] values. The factors associated with worse outcome of tocilizumab use in terms of hospital stay were: baseline status at the start of tocilizumab treatment requiring IMV or supplemental oxygen, elevated levels of ferritin, GOT, GPT, CRP, LDH, lymphopenia, and low PaFi values.

Some studies have investigated factors influencing response to tocilizumab in patients with severe COVID-19. In our case, in contrast to other studies, advanced age and comorbidities were not found to play a crucial role in the lack of efficacy of tocilizumab on mortality [45,46]. In our study, there were no differences in mortality between older and younger patients. In other studies, in contrast to ours, high procalcitonin values seem to predict a worse outcome in tocilizumab patients [45,47]. Procalcitonin is increased in patients with severe COVID-19 infection [48]. In the meta-analysis by Lansbury L et al., a low proportion of patients with COVID-19 were found to have bacterial co-infection [49]. Our findings, as with other studies, suggest that decreased PAFI is associated with worse mortality outcomes, as well as low lymphocyte counts [45,46]. In the study by Sarabia De Ardanaz L et al., it was observed that deceased patients who received tocilizumab, compared to discharged patients, had higher LDH (*p* = 0.013), CRP (*p* = 0.013) and lower lymphocyte (*p* = 0.013) values, as well as a lower mean PAFI index, as in our study [46]. In this study, high neutrophil levels (*p* = 0.024) and low platelet levels (*p* = 0.013) were negative predictors of tocilizumab efficacy. In our study, platelet counts <50,000 cells/mm^3^ were considered exclusion criteria for tocilizumab use. It appears that patients who receive tocilizumab earlier have a lower risk of death [46]. In our study, patients received tocilizumab between 7 and 10 days after the onset of symptoms, that is, during the phase of hyperinflammation and CRS.

Other studies highlight that poor outcomes of tocilizumab use are due to elevated creatinine and CRP values, as well as computed tomography (CT) lung surface area involvement > 50% [45,46]. In our study, all patients fulfilled the latter criterion and CRP was a negative predictor of efficacy in relation to hospital stay.

In our investigation, no pharmacological treatment had sufficient weight in predicting poor outcome with tocilizumab. In other studies, the combination of tocilizumab together with remdesivir did not provide any outcome advantage [45,50]. In our study, all patients took corticosteroids, based on the favourable evidence for the use of corticosteroids in combination with tocilizumab in patients with severe COVID-19 [36,38,45,51]. Thus, the recent meta-analysis by Moosazadeh M confirms that the risk of death in COVID-19 patients treated with corticosteroids and tocilizumab was lower than in the tocilizumab alone and control groups (26% and 52%, respectively) [52].

Our findings confirm that both baseline status at the start of tocilizumab treatment requiring IMV and hyperferritinemia are associated with poor outcome and add insight into predictors of tocilizumab failure in patients with SARS-CoV2 and SRF. This is in line with the Bayesian reanalysis of a previous meta-analysis of 15 studies of hospitalized patients with COVID-19 treated with tocilizumab and corticosteroids [53]. This study showed that simple oxygen therapy and non-invasive ventilation (NIV) was associated with a clinically significant mortality benefit of tocilizumab but not in patients with IMV. Elevated IL-6 levels and hyperferritinemia are considered indicators of systemic inflammation and poor prognosis in COVID-19 [7]. In the study by Lohse A et al., which included 34 patients, those with severity criteria at the start of tocilizumab treatment (high levels of CRP, ferritin, fibrinogen, D-dimer and liver enzymes, as well as lymphopenia) were the ones who would eventually die in the highest proportion [54]. In fact, in this study, as in our case, patients with higher lymphopenia and O_2_ needs and elevated GOT levels had the worst outcome.

On the other hand, the efficacy of tocilizumab treatment seems to be related to the improvement and recovery of the values of analytical parameters associated with systemic inflammation, immune and coagulation disorders, as confirmed by the study by Chamorro-de-Vega E et al., on a Spanish cohort of patients with severe COVID-19 [55]. In that study, the overall mortality was 33.3%, similar to ours, being higher in patients who received IMV than in those who did not (46.2% vs. 26.7%, *p* < 0.001). In our study, the *exitus* rate among IMV patients was 53.8%. A significant improvement in lymphocyte count, CRP, LDH and D-dimer was also observed. These factors are relevant in our ML study as influencing the final objective. Therefore, monitoring these parameters during hospitalisation of the patient on tocilizumab treatment would be the key to see if there is a beneficial response or, on the contrary, an increased risk of mortality. The scientific evidence concludes that a poor outcome is associated with persistently high biomarker values over time after tocilizumab administration, such as LDH, IL-6 and low absolute lymphocyte counts [56]. To circumvent this problem, and due to the large subject-to-subject variability in response to anti-inflammatory treatment for COVID-19, personalized strategies are being developed, guided by rapid cytokine assays, such as the digital immunoassay, which allows rapid “real-time” cytokine monitoring [57].

Nevertheless, a significant number of tocilizumab patients still die regardless of the improvement in some inflammatory biomarkers [55]. Therefore, it is important to consider the timing of IL-6 inhibitor use in relation to disease severity, as it seems to influence treatment efficacy [58]. In our case, tocilizumab was administered within 7–10 days of the first symptoms, that is, during the hyperinflammation phase. It has been suggested that administration of treatment in critical illness may not reverse cytokine-mediated injury that has already occurred [59]. Deaths in tocilizumab-treated patients may be due to patients with a poor initial prognosis and a potentially irreversible hyperinflammatory syndrome [60].

There is evidence for the existence of different phenotypes of COVID-19 patients with different inflammatory and immune response, mortality risk and response to pharmacological treatment and respiratory support [32,61,62]. In the study by Chen H et al., using an ML approach, two phenotypes of COVID-19 were identified: hypoinflammatory and hyperinflammatory, the latter characteried by elevated levels of proinflammatory cytokines and higher rates of complications [62]. Steroid treatment was associated with reduced 28-day mortality (HR, 0.45; 95% CI, 0.25–0.80; *p* = 0.0062) in the hyperinflammatory phenotype. Treatment failure with tocilizumab may support this hypothesis, raising the need to identify factors to tailor treatments.

Our study has limitations and strengths. The main limitations are secondary to the retrospective methodology and the single-centre design with a limited sample, but these are compensated for by the use of powerful tools such as ML. In addition, one of the main advantages of the ML methods is the used of small datasets. Thus, the complexity and classification time of the method proposed can be very low. This is the main reason of using just one small dataset. Moreover, data augmentation technique has been employed [26]. Another limitation is that during the period in which the study was conducted, patients were not yet optimally treated, and this could somewhat influence the results of the study and add confounding factors to our results.

Among the strengths is the existence of a diverse patient population with a relatively high prevalence of comorbidities, which tend to be under-represented in most RCTs. As such, our results are more generalizable. This methodology has the ability to identify patients in whom tocilizumab treatment is associated with longer survival time. Lam C et al., used the ML system to identify hospitalized patients with COVID-19 in whom treatment with a corticosteroid or remdesivir is associated with increased survival time [63]. Similarly, ML algorithms have been used to predict mortality in COVID-19 patients treated with corticosteroids and remdesivir [64]. Comparative studies have revealed that ML methods can be more accurate and efficient than traditional logistic regression analysis, especially when sample size is limited [65].

## 5. Conclusions

Tocilizumab has been shown to be beneficial in patients with SRF associated with SARS-CoV-2 pneumonia. However, a non-negligible number of patients die or require longer hospital stays until recovery or *exitus*, despite treatment with tocilizumab. Therefore, we use ML techniques, which are the most sophisticated and accurate tools for predicting events of interest. With them, we selected predictors of tocilizumab treatment failure in patients with severe COVID-19. XGB was the method with the highest predictive accuracy. Among these factors, patients with baseline IMV at the start of tocilizumab treatment and/or hyperferritinemia were candidates for higher tocilizumab treatment failure. This should be taken into account in clinical practice to identify these patients as they will require more aggressive management and closer follow-up to avoid a poor outcome. Further research is needed to advance the understanding of pathophysiological mechanisms and characterize immunoinflammatory alterations across the full range of phenotypes to better classify patients for optimal clinical management. These ML methods will help improve patient outcomes and resource allocation during the COVID-19 pandemic.

## Figures and Tables

**Figure 1 jcm-11-04729-f001:**
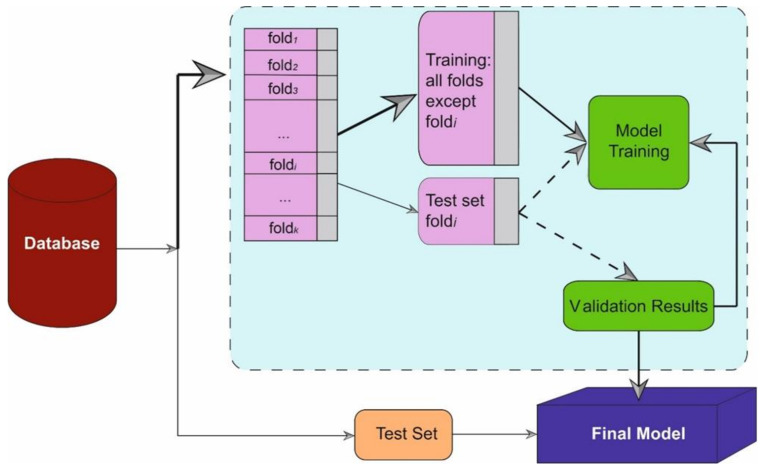
The figure shows the scheme followed in the learning and testing process of this work.

**Figure 2 jcm-11-04729-f002:**
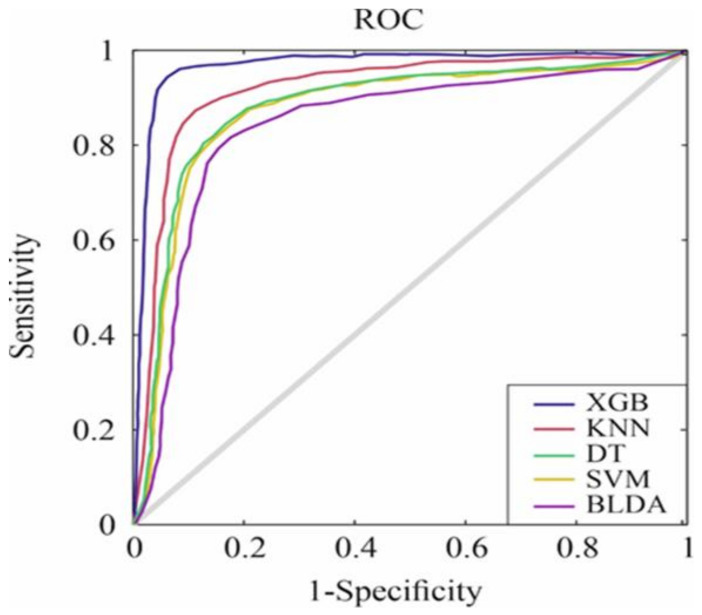
ROC curves for the five assessed machine learning predictors. Abbreviations: ROC: receiver operating characteristic, XGB: extreme gradient boost, KNN: K-nearest neighbour, DT: decision tree, SVM: support vector machine, BLDA: Bayesian linear discriminant analysis.

**Figure 3 jcm-11-04729-f003:**
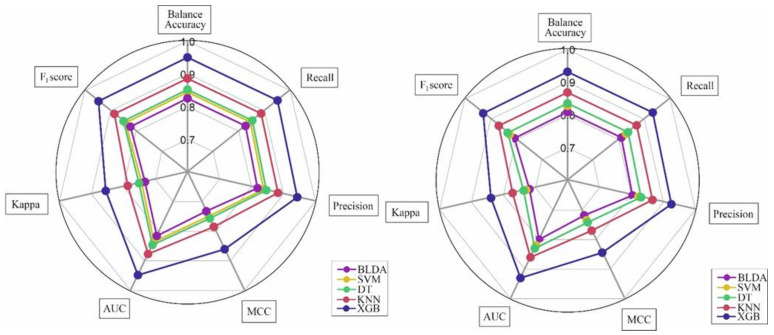
The figure shows the radar plot of the training phase (**left**) and validation (**right**) for the prediction of mortality in COVID-19 patients treat to tocilizumab. Abbreviations: AUC: area under curve, MCC: Matthew’s correlation coefficient, XGB: extreme gradient boost, KNN: K-nearest neighbour, DT: decision tree, SVM: support vector machine, BLDA: Bayesian linear discriminant analysis.

**Table 1 jcm-11-04729-t001:** Summary of the mean values and standard deviation of balanced accuracy, recall, precision, *F*_1_ score, AUC, MCC, DYI and Kappa index of the machine learning models and the proposed method implemented in this article.

Methods	Balanced Accuracy	Recall	Precision	*F*_1_ Score
SVM	82.77 ± 0.47	82.87 ± 0.53	82.18 ± 0.57	82.52 ± 0.62
BLDA	80.91 ± 0.81	81.02 ± 0.78	80.31 ± 0.75	80.66 ± 0.73
DT	83.54 ± 0.63	83.63 ± 0.68	83.01 ± 0.64	83.32 ± 0.62
KNN	86.86 ± 0.54	86.98 ± 0.51	86.59 ± 0.46	86.79 ± 0.47
XGB	93.16 ± 0.25	93.25 ± 0.31	92.49 ± 0.27	92.87 ± 0.29
**Methods**	**AUC**	**MCC**	**DYI**	**Kappa**
SVM	0.82 ± 0.02	73.45 ± 0.58	82.78 ± 0.61	72.91 ± 0.59
BLDA	0.80 ± 0.02	71.79 ± 0.73	80.92 ± 0.72	71.89 ± 0.71
DT	0.83 ± 0.02	74.18 ± 0.67	83.55 ± 0.65	73.71 ± 0.68
KNN	0.86 ± 0.02	76.97 ± 0.48	86.87 ± 0.45	77.15 ± 0.46
XGB	0.93 ± 0.02	84.41 ± 0.25	93.17 ± 0.26	83.91 ± 0.28

Abbreviations: SVM: support vector machine, BLDA: Bayesian linear discriminant analysis, DT: decision tree, KNN: K-nearest neighbour, XGB: extreme gradient boost, AUC: area under curve, MCC: Matthew’s correlation coefficient, DYI: degenerated Younden index.

**Table 2 jcm-11-04729-t002:** Parameters used for the implementation of different machine learning methods.

Method	
SVM	C = 1.0
	sigma = 0.5
	Numerical tolerance = 0.001
	Iteration limit = 100
	Kernel function: Linear kernel, Gaussian, Quadratic and Cubic
BLDA	Kernel: Bayesian
DT	Minimum number of instances in leaves = 4
	Minimum number of instances in internal nodes = 6
	Maximum depth = 100
KNN	Number of neighbours = 20
	Distance metric: Euclidean
	Weight: Uniform
XGB	Base estimator: tree
	Maximum number of splits = 20
	Learning rate = 0.1
	Number of learners = 50

Abbreviations: SVM: support vector machine, BLDA: Bayesian linear discriminant analysis, DT: decision tree, KNN: K-nearest neighbour, XGB: extreme gradient boost.

**Table 3 jcm-11-04729-t003:** Basal clinical data of patients. Data are *n* (%) or median (IQR), unless otherwise stated.

Variable	Cohort
Number of patients	67
Age (years) (IQR)	65 (57–74.5)
Male (%)	43 (64.2)
Exitus, *n* (yes %)	24 (35.8)
Hospital admission (days) after tocilizumab administration (IQR)	14 (10–29.5)
IMV, *n* (yes %)	13 (19.4)
7-day mortality, *n* (yes %)	7 (10.4)
21-day mortality, *n* (yes %)	10 (14.9)
Antivirals drugs, *n* (yes %)	30 (44.8)
Lopinavir/ritonavir, *n* (yes %)	27 (40.3)
Remdesivir, *n* (yes %)	3 (4.5)
Hydroxychloroquine, *n* (yes %)	34 (50.7)
Interferon-beta, *n* (yes %)	7 (10.4)
Anakinra, *n* (yes %)	33 (49.2)
Baseline situation at the start of tocilizumab treatment requiring supplemental oxygen, *n* (yes %)	60 (89.5)
Baseline situation at the start of tocilizumab treatment requiring IMV, *n* (yes %)	7 (10.4)
Smoker/ex-smoker, *n* (yes %)	14 (20.9)
Diabetes, *n* (yes %)	18 (26.9)
COPD, *n* (yes %)	4 (5.9)
Arterial hypertension, *n* (yes %)	34 (50.7)
Dyslipemia, *n* (yes %)	21 (31.3)
Obesity [BMI ≥ 30 kg/m^2^], *n* (yes %)	4 (5.9)
Ischemic heart disease, *n* (yes %)	5 (7.5)
Chronic kidney disease, *n* (yes %)	2 (2.9)
Lymphocytes (10 × 9/L) (IQR)	0.95 (0.5–1.3)
CRP (mg/L) (IQR)	12.5 (5.8–21.1)
LDH (U/L) (IQR)	759 (538–934)
Procalcitonin (ng/mL) (IQR)	3.49 (0.3–10.9)
Ferritin (µg/L) (IQR)	828 (412–1388)
FiO_2_ (%) (IQR)	37.5 (28–50)
PaFi (IQR)	198.5 (135–251.5)
GPT (U/L) (IQR)	39 (26–66.5)
GOT(U/L) (IQR)	45 (31–67)

Abbreviations: IQR: interquartile range, IMV: invasive mechanical ventilation, COPD: chronic obstructive pulmonary disease, BMI: body mass index, CRP: C-reactive protein, LDH: lactate dehydrogenase, FiO_2_: inspired oxygen fraction, PaFi: ratio between arterial oxygen pressure and inspired oxygen fraction (PaO_2_/FiO_2_), GPT: glutamate-pyruvate transaminase, GOT: glutamate-oxaloacetate transaminase.

## Data Availability

The datasets used and/or analyzed during the present study are available from the corresponding author on reasonable request.

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
