# Peer review of "Application of Machine Learning in Hospitalized Patients with Severe COVID-19 Treated with Tocilizumab"

_jcm, 2022, doi:10.3390/jcm11164729_

Round 1
Reviewer 1 Report
The authors sought to develop and compare the ability of various machine learning techniques to predict poor outcomes among patients with COVID-19 who receive tocilizumab.
1. It was unclear to me if there were separate training and validation cohorts. I did not see mention of a validation cohort, but the results section mentions validation in line 222.
2. The factors associated with worse outcomes among those who receive tocilizumab are likely similar to the factors that generally predict poor outcomes. It would be more clinically useful to know whether tocilizumab still benefits these patients, when comparing them to those who did not receive tocilizumab.
3. The results section does not include information about the models. In the abstract, the authors mention factors associated with worse outcomes, but these data are not presented in the results. It isn’t until the second page of the discussion that these factors are mentioned. As a result, there seems to be a disconnect between the abstract/intro and results. The way it is currently written, the results section seems more about various machine learning techniques than tocilizumab in COVID.
4. In the conclusion the authors state “Thus, in contrast to other studies, in our case, advanced age and comorbidities were not found to play a crucial role in the lack of efficacy of tocilizumab on mortality.” (lines 368-370). Since there is no control group, I am not sure you can comment on efficacy. You could state mortality was similar between older and younger patients, but I am not sure you have the power to really comment on this.
5. The limitation section (starting on line 446, page 10) requires revision. Machine learning cannot compensate for small sample size or for a single center study. In fact, these factors may be quite problematic for machine learning techniques. Additionally, there are limitations to machine learning approaches which have not been discussed.
6. There is no description of the 67 patients in the study. We do not know what the male/female breakdown is, the median age, timing of tocilizumab, etc.
Author Response
Response to Reviewer 1 Comments:
"Please see the attachment."

Reviewer 2 Report
I do not have a background in modeling and cannot comment on the use of machine learning models but am able to comment on study population, the data used, clinical relevance of the data and the significance of the outcomes.
Tocilizumab use in patients with severe SARS-CoV-2 that is rapidly progressing is an extremely important topic and there are gaps in the current literature identifying the patient population in which therapy will provide the most benefit. As the authors conclude, once you are already on IMV, it is already likely too late to have a benefit. The remainder of their conclusions are difficult to interpret likely because of the time period of study early in the pandemic (march 2020-september 2020) when the use of therapeutics was evolving and different patient approaches were used week to week depending on the data coming out in the literature and the international discussions on approach.
Introduction - page 1, line 38. I question how a silent viral replication phase could include cough, fever and dyspnea.
Study population - The hospital's internal protocol, requiring symptom onset of > 7 days to administer tocilizumab. I believe a large amount of patients that may have clinically benefitted was missed and the outcomes may be shifted to patients with poor outcomes when compared to other studies. I question the turn-around time on IL-6 to use as a marker to include in protocol, but potentially was fast at this center; please comment.
Listed as exclusion criteria for toci therapy is presence of comorbidity that could lead to a poor prognosis based on clinical judgement. I believe this needs to be fleshed out more - was there a list of criteria, was more than one person making these decisions that would lead to variability? were these mostly immunocompromised hosts? Did they overlap with the same criteria that would lead someone to be more likely to have severe disease? I do realize this may be difficult as provider decision-making would have evolved over this time period.
There were many therapies used during the course of this study as at any institution. Because I am not well versed in the machine learning models, I may not have the correct interpretation. It appears that no agent was used enough to give enough weight to the model but many used have been shown not to be beneficial (lopinavir/ritonavir, HCQ) but do have significant side effects (Qtc prolongation with azithro/HCQ). This seems to not be relevant though because none was used to a great extent in these 67 patients.
Generalizability of study - The only conclusion I am able to make from this study is that toci likely provides little benefit in patients in the inflammatory phase of COVID-19 after they have already been placed on invasive ventilation. This is well known in the literature. The other variables are difficult to make conclusions on without knowing values or timing of laboratory results, we would expect that many of the acute phase reactants be elevated during the inflammatory phase of COVID-19, elevated liver enzymes in patients suspected to be in shock (as per inclusion criteria) and lymphopenia is present in many patients even with mild SARS-CoV2 disease.
I do believe that there is value in use of machine learning to potentially identify the ideal patient population for tocilizumab therapy but do not believe that this study helps reinforce or change my approach in this patient population.
Author Response
Response to Reviewer 2 Comments
"Please see the attachment".

Round 2
Reviewer 1 Report
Rephrase “Similarly, baseline treatment contemplated the use of azithromycin 500..” I don’t think “contemplated” is the word you want. Methods line 130
Methods line 146: Consider “tocilizumab initiation” rather than “tocilizumab starting therapy”
Table 3: rather than “Hidroxicloroquina” change to “hydroxychloroquine”
Regarding prior comments:
1. It is now clear that there was a validation cohort.
2. I understand the author's response. Clinically, it would be more useful to know whether the tocilizumab improves outcomes rather than simply prognosticating for those who do receive tocilizumab, but that may be beyond the scope.
3. The authors now include the model information in the results which makes it clearer.
4. Would be helpful for the authors to comment on why their study differed from the other studies rather than just mentioning that the results are different.
5. I am not an expert in machine learning so would need a statistician to review.
6. Table 3 is helpful for understanding the population.
Author Response
Response to Reviewer 1 Comments
"Please see the attachment."
